# Rapid and Specific Action of Methylene Blue against *Plasmodium* Transmission Stages

**DOI:** 10.3390/pharmaceutics14122794

**Published:** 2022-12-14

**Authors:** Nathanaël Saison, Jean-François Franetich, Yudi T. Pinilla, Anton Hoffmann, Stravensky T. Boussougou-Sambe, Barclaye Ngossanga, Maurel Tefit, Kutub Ashraf, Nadia Amanzougaghene, Shahin Tajeri, Ayola A. Adegnika, Dominique Mazier, Steffen Borrmann

**Affiliations:** 1Centre de Recherches Médicales de Lambaréné, Lambaréné 1437, Gabon; 2Institute for Tropical Medicine, University of Tübingen, 72074 Tübingen, Germany; 3Centre d’Immunologie et des Maladies Infectieuses, Inserm, CNRS, Sorbonne Université, U1135, ERL8255, CIMI-Paris, F-75013 Paris, France; 4German Center for Infection Research (DZIF), Partner Site Tübingen, 38124 Braunschweig, Germany

**Keywords:** *Plasmodium falciparum*, *Plasmodium berghei*, *Plasmodium yoelii*, *Anopheles gambiae*, *Anopheles stephensi*, methylene blue, Gabon, transmission, in vivo and ex vivo experiments

## Abstract

Methylene blue (MB) is the oldest synthetic anti-infective. Its high potency against asexual and sexual stages of malaria parasites is well documented. This study aimed to investigate possible additional activities of MB in interfering with parasite transmission and determine target stages in *Anopheles* vectors and humans. MB’s transmission-blocking activity was first evaluated by an ex vivo direct membrane feeding assay (DMFA) using *Plasmodium falciparum* field isolates. To investigate anti-mosquito stage activity, *Plasmodium berghei*-infected *Anopheles stephensi* mosquitoes were fed a second blood meal on mice that had been treated with methylene blue, 3, 6- and 15-days after the initial infectious blood meal. Anti-sporozoite and liver stage activities were evaluated in vitro and in vivo via sporozoite invasion and liver stage development assays, respectively. MB exhibited a robust inhibition of *P. falciparum* transmission in *An. gambiae*, even when added shortly before the DMFA but only a moderate effect against *P. berghei* oocyst development. Exposure of mature *P. berghei* and *P. falciparum* sporozoites to MB blocked hepatocyte invasion, yet *P. berghei* liver stage development was unaffected by MB. Our results indicate previously underappreciated rapid specific activities of methylene blue against Plasmodium transmission stages, preventing the establishment of both mosquito midgut and liver infections as the first essential steps in both hosts.

## 1. Introduction

The development of new interventions that target transmissible stages of malaria parasites will play a key role in the goal of malaria elimination. Existing conventional control tools such as insecticide-treated bed nets, prompt and efficacious treatment and vector population reduction led to a significant reduction of the malaria burden. Still, malaria remains an important cause of morbidity, especially among African children under 5 years of age. In 2019, the WHO estimated even a notable surge in morbidity and mortality after two decades of steady decreases (WHO World Malaria Report 2021) [1]. Despite the worrying emergence of parasite resistance to both artemisinin and partner drugs [2,3], ACTs continue to be highly efficacious in treating uncomplicated *P. falciparum* malaria in African countries [4,5]. Artemisinin derivatives lead to rapid asexual parasite clearance in sensitive infections [6] and are also highly active against stage I-III sexual transmission stages (gametocytes). However, artemisinins show only incomplete activity against stage IV and V gametocytes [7]. Therefore, gametocytes continue to circulate after ACT treatment, and the transmission chain is not disrupted [8]. High in vivo gametocidal activity against mature *P. falciparum* gametocytes has so far only been confirmed for the 8-aminoquinolines primaquine and tafenoquine and methylene blue [7,9,10]. The WHO recommends adding a single low dose of primaquine (PQ; 0.25 mg base/kg) in combination with ACT treatment of uncomplicated *P. falciparum* malaria [11,12]. However, the implementation of PQ in endemic areas remains difficult because of concerns about the risk of hemolytic anemia (AHA) in individuals with phosphate dehydrogenase (G6PD) deficiency, an inherited X chromosome-linked abnormality. The prevalence of G6PD deficiency varies between 5 and 30% in malaria-endemic areas of Asia and Africa [11,12].

Methylene blue (MB), also known as methylthioninium chloride, is the oldest synthetic anti-infective. Its first successful use for the treatment of malaria was reported in 1891 [13]. MB remains active also against artemisinin-resistant parasites [14,15]. Despite good efficacy, the clinical importance of MB for the treatment of malaria was relatively quickly curtailed by two prominent, though reversible, side effects: urine and sclera discoloration [16]. MB treatment is also associated with a slight reduction of hemoglobin concentrations, but there is no evidence of an excess of severe hemolytic anemia [17,18]. The mode of action of MB against *Plasmodium* is incompletely defined, but the inhibition of (met)hemoglobin degradation appears to play a key role (reviewed in Schirmer et al. [16]). MB has also been shown to be involved as a direct inhibitor of glutathione reductase (GR), a vital enzyme of the parasite, protecting it against oxidative stress [19]. MB acts as a redox cycling agent in a complex catalytic cycle leading to the production of reactive oxygen species (ROS), which are toxic and affect the survival of the living parasite [20].

Moreover, MB has been shown to be active against *Plasmodium* spp. gametocytes [7,21]. Indeed, previous studies reported a morphological deformation on *P. falciparum* gametocytes MB treatment [22] and a sterilizing effect [23] with a preferential clearance for male gametocytes [10], which reduces the gametocyte infectivity and blocks the transmission of *P. falciparum* from the human to the mosquito. Oral MB treatment combined with ACTs has been shown to be safe and efficacious in the treatment of uncomplicated *P. falciparum* malaria in Mali and in Burkina Faso [10,24]. These studies demonstrate that MB could be a potential additional tool for reducing malaria transmission in endemic areas. Similar studies using a standard membrane feeding Assay (SMFA) have investigated the transmission-blocking potency of MB, but none of them were under field conditions [7]. Furthermore, there are no studies in the literature relating an effect on *Plasmodium* development in the vector. The sporontocidal effect of MB within the vector, as well as its effect on the *Plasmodium* sporozoite stage, thus still remained to be determined

In this context, this study aimed to evaluate the ex vivo *P. falciparum* transmission-blocking activity of MB in *An. gambiae* mosquitoes using fresh parasite isolates obtained from symptomatic and asymptomatic individuals with microscopic evidence of *P. falciparum* gametocytes. We then evaluated the activity of MB against oocysts, sporozoites and *Plasmodium* liver stages in murine malaria models using *P. berghei* and *P. yoelii*.

## 2. Materials and Methods

### 2.1. Sample Collection for Ex Vivo Study

The study was conducted at the Centre de Recherches Médicales de Lambaréné (CERMEL), Gabon, from September 2019 to March 2020. Malaria transmission in this hyperendemic area is considered perennial [25]. Participants in the rural area surrounding Lambaréné were screened for the presence of *P. falciparum* gametocyte-positive infections. Finger-prick blood was collected and used for the preparation of thick smears from volunteers aged ≥1 year after signed informed consent of participants or caretakers. Asymptomatic and symptomatic participants with microscopic evidence of *P. falciparum* gametocytes, aged ≥5 years and with a hemoglobin concentration of ≥6 g/dL, were selected as blood donors for the direct membrane feeding assay (DMFA). The following day of the screening, 5 mL of venous blood was collected into a heparinized tube. All participants with confirmed malaria infection were treated either on the following day by the village health workers or following the venous blood collection at the laboratory with artemether-lumefantrine (Coartem, Novartis, Basel, Switzerland) twice daily for 3 days according to manufacturer’s instructions. The protocol of this study was approved by the institutional review board of CERMEL (CEI 013/2019).

### 2.2. Mosquito Species

Two species of *Anopheles* were used in this study:*Anopheles gambiae* s. s. (Kisumu strain) were reared at the Laboratory of Entomology at CERMEL in Lambaréné (Gabon). Adult mosquitoes were kept at standard insectary conditions (26 ± 0.5 °C and 80% ± 10 relative humidity) in heating cabinets with a 12:12 h (h) light and dark cycle. They were fed with a 10% sugar solution. Larvae were hatched in room temperature water and fed with a small amount of Vitakraft^®^ Premium VITA Flake-Mix fish provided daily;*Anopheles stephensi* were reared at the Insectary at the Center for Immunology and Infectious Diseases (CIMI) in Paris under similar conditions.

### 2.3. Parasites

Three species of *Plasmodium* were used in in vivo and in vitro experiments:*Plasmodium berghei* ANKA GFP, a parasite line constitutively expressing GFP as previously described [26];*Plasmodium yoelii* (17XNL strain), a parasite line expressing both GFP and Luciferase reporters (GFP-luc) as previously described [26];*Plasmodium falciparum* (NF54 strain) sporozoites were obtained from infected salivary glands of *A. stephensi* 14–21 days after an infective blood meal (Department of Medical Microbiology, Radboud University Medical Center, Nijmegen, The Netherlands).

### 2.4. Methylene Blue

For our ex vivo assay, MB was obtained from Sigma-Aldrich, St. Louis, MO, USA and stocked solutions were prepared at 2.5 mg/mL in distilled water. For our in vivo and in vitro assays, Provedye^®^, a pharmaceutical form of MB kindly provided by Provepharm, was used.

### 2.5. Ex Vivo Transmission Blocking Assessment of MB

All ex vivo transmission procedures were performed in the BSL-3 area of the insectary to prevent the release of *Plasmodium*-infected mosquitoes. Three to 6-day old adult *Anopheles gambiae* Kisumu strain mosquitoes were used for the ex vivo direct membrane feeding assay. The heparinized blood collected from participants was centrifuged for 5 min at 2000 rpm at 37 °C. Plasma was removed, and packed red blood cells (RBCs) were washed with RPMI 1640 medium, repeated twice, and reconstituted to 40% hematocrit with non-immune human AB serum. MB Sigma-Aldrich, USA, dissolved in distilled water, was added <5 min to the prepared blood before transferring the isolate to the BSL-3 area for the DMFA. Two aliquots with an MB concentration of 5 µM and 10 µM were then prepared. Fifty 12 h-starved female mosquitoes per paper cup were allowed to take a blood meal for 30 min in the dark via an artificial membrane attached to glass feeders kept at 37 °C (Parafilm^®^). Blood-fed mosquitoes were kept on a 10% sucrose diet at standard insectary conditions, and mosquito mortality was reported and monitored until the midgut dissection. Unfed mosquitoes were discarded. Mosquito midguts were dissected 6–9 days post-infection, stained with 1.2% mercurochrome and visualized under a light microscope at 100×. The infection rate was calculated by dividing the number of infected dissected mosquitoes by the total number of dissected mosquitoes. Infection intensity was calculated as the mean number of oocysts in all dissected mosquitoes.

### 2.6. Assessment of Anti-Mosquito Stage Activity of MB by Secondary In Vivo Feeding

The *P. berghei* ANKA GFP line, a parasite line constitutively expressing GFP as previously described [26], was preliminarily maintained in mice (BALB/c). Mice were infected by intra-peritoneal (i.p.) inoculation of 10^7^ erythrocytes parasitized with *P. berghei* from cryopreserved parasite stocks. On day 3, the presence of gametocytes was assessed, and mice were anesthetized and placed on the top of individual cages containing starved female *An. stephensi* mosquitoes, which were allowed to feed for 30–60 min. Three, 6- and 15-days after the infectious blood meal, around 50 infected mosquitoes were fed on uninfected mice that were either untreated or treated with MB (50 mg/kg and 100 mg/kg) 1 h before with an i.p. inoculation of a single dose of Provedye^®^, a pharmaceutical form of MB kindly provided by Provepharm. Unfed mosquitoes were removed and discarded. Fed mosquitoes were maintained in climatic chambers at standard mosquito-rearing conditions. Between 9 and 11 days after infections, 30 mosquitoes per treatment group were dissected. Midguts were examined by fluorescent microscopy at 400× magnification. Images of the whole midguts were captured in both brightfield and fluorescence modes (Leica DMI 4000 microscope equipped with an MRC5 Zeiss camera). The oocyst burden was determined by counting the number of oocysts using ImageJ image analysis software [27]. For mosquitoes that had been fed a second time on day 15 after the infectious blood meal, salivary glands were dissected on day 21 after the infectious blood meal and sporozoites were counted using a Neubauer chamber at 400× magnification.

### 2.7. In Vitro Sporozoite Invasion Assay

Primary simian and human hepatocytes (90.000 cells per well) were seeded into collagen-coated black 96-wells plates and maintained at 37 °C in 5% CO_2_ in complete medium (William’s E medium supplemented with 10% of fetal clone III serum, 1% penicillin-streptomycin, 5 × 10^−3^ g/L human insulin, 5 × 10^−5^ M hydrocortisone, 1/70 Matrigel). *P. falciparum* and *P. berghei* sporozoites were used for the invasion assays. Between 18–21 days post-infection, salivary glands of infected mosquitoes were isolated by hand dissection, sporozoites were counted, resuspended and aliquoted into Eppendorf tubes to a final concentration of 20,000 sporozoites/50 µL in phosphate-buffered saline (Gibco, Life Technologies, Carlsbad, CA, USA). MB, dissolved in distilled water, was added to the sporozoite suspensions and mixed gently. Final MB concentrations ranged between 1.25–50 µM. Aliquots were incubated at room temperature for 1 h and then centrifuged for 5 min at 4000× *g* to remove MB from the suspension. The sporozoites were resuspended in a complete medium. This washing step was repeated to remove residual MB concentrations. Sporozoites were then added to the hepatocyte cultures (20.000 sporozoites/well). Each MB concentration was tested in quadruplicates. The infected hepatocyte culture plates were centrifuged for 10 min at 750× *g* at room temperature, allowing fast parasite sedimentation and then incubated for 3 h at 37 °C and 5% CO_2_ to promote parasite invasion of hepatocytes. Extracellular sporozoites were then washed away, and plates were returned to the incubator with fresh media. Cells were fixed with cold methanol at 48 h post-infection (pi) for *P. berghei* and at 6 days pi for *P. falciparum. P. falciparum* exo-erythrocytic forms (EEFs) were immune-stained using a mouse polyclonal serum raised against the PfHSP70 and an Alexa-Fluor 488 conjugate anti-mouse antibody. All nuclei were stained with DAPI. The anti-malarial activity was evaluated by counting and sizing EEFs (Figure A3) using a Cell-Insight High Content Screening platform equipped with the Studio HCS software (ThermoFisher Scientific, Waltham, MA, USA) at Paris Brain Institute. Cell cytotoxicity was evaluated by assessing cell confluence and by comparing the numbers of DAPI-positive hepatocytes before vs. after drug treatment.

### 2.8. In Vitro Liver Stage Development Assay

Primary simian hepatocytes (90.000 cells per well) seeded into collagen-coated 96-wells plates were maintained at 37 °C in 5% CO_2_ in a complete medium. 3 × 10^4^ freshly dissected sporozoites of *P. berghei* were resuspended in complete medium and added to each well. Infected hepatocytes were centrifuged for 10 min at 750× *g* and further incubated for 2 h as above. MB treatment of infected cultures was initiated at different time points, simultaneously, i.e., during the infection, 2 h pi and 12 h pi and renewed for all cultures at 24 h pi. Final MB concentrations ranged between 1.25–100 µM. Parasite numbers and sizes were determined using the same procedure as described above for the in vitro sporozoite invasion assay.

### 2.9. In Vivo Sporozoite Invasion/Development Assay

Six weeks old female BALB/C mice were randomly allocated into four groups of 5 mice. Two groups were treated with MB at 50 mg/kg by i.p. injection. Untreated mice (2 groups of 5 mice) were used as a negative control. MB was administrated 30–40 min before mosquito feeding or sporozoite inoculation in order to reach peak plasma concentrations of MB during the short intra-vascular travel of sporozoites to the liver. One MB-treated group and one control group were challenged by retro-orbital injection of 5000 *P. yoelii* (GFP-luc strain) sporozoites (17XNL strain), a parasite line expressing both GFP and Luciferase reporters (GFP-luc) as previously described [26], while the last two groups were infected by the bite of 20 *P. berghei*-infected mosquitoes per mice. Liver and blood stage development was monitored at 44 h pi by bioluminescent imaging. Luciferase activity was monitored using an intensified charged-coupled device video camera of the In Vivo Imaging System (IVIS, Caliper Life Science, Hanover, MD, USA). 10 min before bioluminescence signal acquisition, 100 µL of luciferin sodium salt dissolved in phosphate-buffered saline (100 mg/kg) was ip-administered, and mice were anesthetized with isoflurane. They were then placed into the camera chamber, and bioluminescence imaging was acquired. Images were acquired and analyzed using the Living Image 3.0 software (Caliper Life Science, Hanover, MD, USA). To assess blood stage development in *P. berghei*-infected mice, the presence of parasites and parasitemia were determined by counting *P. berghei* on blood smears at days 6 and 7 pi.

### 2.10. Statistical Analysis

Data were entered into an Excel spreadsheet (Microsoft Office 2018) before being analyzed with Prism 7.03. (GraphPad, San Diego, CA, USA). Differences in oocyst prevalence (number of oocyst-positive mosquitoes per group) were examined using a binomial distribution for the number of positive mosquitoes. A zero-inflated negative binomial distribution was adjusted to compare the proportion of oocysts per mosquito (including negative insects) [28]. The bootstrapping process was used to get 95% confidence interval values. We set the maximal inhibition to 100%, and the minimum inhibition (zero drug concentration) and the drug concentration producing 50% of the maximum effect (IC_50_) were estimated. Statistical significance was defined as a *p*-value of 0.05 or less.

## 3. Results

### 3.1. Short-Term Exposure of Fresh P. falciparum Isolates to MB Efficiently Blocks Mosquito Infections

Using DMFAs with 5 fresh *P. falciparum* gametocyte-containing patient isolates, we established a robust inhibition of *P. falciparum* infection after only short-term incubation with MB (<5 min before feeding) in laboratory-reared An. gambiae Kisumu strain colonies (Figure 1). The gametocyte densities of *P. falciparum* in the 5 isolates ranged from 212/µL to 716/µL (median 288) (Table A1, Appendix A). We evaluated the effect of 2 concentrations of MB (5 and 10 µM) on *P. falciparum* infection prevalence, oocyst density and An. gambiae mortality. There were significant differences between the MB treatment groups for infection prevalence (F (2,12) = 232 *p* < 0.0001) compared to the control group. Mean *P. falciparum* infection prevalence was significantly reduced in mosquitoes that ingested MB at 10 µM (90.3%; 95% CI 86.0–94.1; *p* < 0.0001 compared to control), and at 5 µM (78.7%; 95% CI 74.1–82.0; *p* < 0.0001 compared to control). Mean infection intensity (i.e., the number of oocysts per mosquito) was reduced in the groups of mosquitoes that ingested MB-exposed infective blood meals by 98.8 (95% CI 96.2–99.9 *p* < 0.001) for 10 µM vs. control and by 96.0 (95% CI 85.4–100 *p* < 0.0001) for 5 µM vs. control. Mosquito mortality rates were similar in the 10 µM MB group (19.8%; *p* > 0.3), in the 5 MB µM group (16.2%; *p* > 0.9] compared to the control group (16.3%) (Appendix A, Figure A1).

### 3.2. Limited Effect of MB on Plasmodium berghei Mosquito Stage Development

In mosquitoes fed with an MB containing secondary blood meal on day 3 after being infected with *P. berghei*, the mean prevalence of oocysts in midguts dropped from 84% in the control group (total of 36 infected of 43 mosquitoes) to 54% (total of 22 infected of 41 mosquitoes; *p* < 0.001) and 76% (total of 31 infected of 41 mosquitoes; *p* = 0.4) in respectively the 100 and 50 mg/kg MB groups (Table A2, Appendix B). Likewise, infection intensity (Figure 2) was reduced by the secondary blood meal containing MB (median 27 for 100 mg/mL and 74 oocysts per mosquito 50 mg/mL) compared to control (median 97 oocysts per mosquito; *p* < 0.001 for comparisons of MB 100 mg/mL vs. control). In terms of the prevalence of infection and infection intensity in mosquitoes fed on day 6 post-infection, there was no significant difference between MB-treated groups vs. control (Table A2, Appendix B). No significant difference was seen in the number of salivary gland sporozoites in MB-treated groups vs. control (Table A3, Appendix B). In the mosquitoes fed with a secondary MB-containing blood meal on day 15 post-first infection, no statistically significant effect on sporozoite numbers per mosquito versus control was observed (Table A3, Appendix B).

### 3.3. Pre-Exposure of Sporozoites with MB Inhibits Hepatocyte Invasion

Freshly dissected sporozoites were incubated for 30 min with MB concentrations ranging from 1.25 to 50 µM. The effect of drug exposure on sporozoite invasion was evaluated by counting the number of EEFs 48 h or 5 days after inoculation of the hepatocytes with the pre-incubated sporozoites of *P. berghei* and *P. falciparum*, respectively. Notably, MB exhibited significant hepatocyte invasion-blocking activity (IC_50_ = 4.46 µM for *P. berghei*; IC_50_ = 4.7 µM for *P. falciparum*). No significant cytotoxicity towards hepatocytes was observed compared to the drug-free control as measured by the number of hepatocytes nuclei (Figure 3).

### 3.4. MB Inhibits Early In Vitro Liver Stage Development

MB activity against *P. berghei* liver stages was investigated at concentrations ranging from 1.25 to 100 µM. MB was either added simultaneously to the hepatocyte invasion or at 2 h and 12 h pi until parasite maturation at 48 h pi. Interestingly, we found that MB had moderate activity on *P. berghei* liver stage development, with a more pronounced effect when MB was added at the same time as the parasites’ invasion (IC_50_ = 13.9 µM, 33.6 µM and 34.7 µM at 0 h, 2 h or 12 h pi, respectively (Figure 4). Moreover, MB also showed an effect on the parasite development as evidenced by the reduction of *P. berghei* EEFs size (Appendix B, Figure A2).

### 3.5. Reduction of In Vivo Liver Stage Development by MB

Liver parasite load was significantly reduced in mice inoculated intravenously with *P. yoelii* sporozoites 30 min after 50 mg/kg MB treatment. However, the complete blockade of parasite development was not achieved (Figure 5). The mean luminescence (photons/sec) values measured at liver positions of mice treated with MB at 44 h pi was 4.7 × 10^5^ (SD, 4.6 × 10^5^) vs. 1.9 × 10^6^ (SD, 3.5 × 10^6^) for the untreated control group. In the *P. berghei* groups (Figure 6) challenged by mosquito bite, blood stage parasitemia was significantly reduced in the MB group [mean parasitemia 0.112 ± 0.105, *p* = 0.0159 by comparison vs. control] at day 6 compared to the control [mean parasitemia 0.564 ± 0.242]. These results indicate that MB is moderately effective in preventing rodent malaria parasite liver and subsequent blood-stage infections.

## 4. Discussion

Recent investigations on the antimalarial activity of MB are rekindling the debate on its role in improved malaria control and, eventually, eradication [14]. The WHO called for more research on the transmission-blocking potential of MB in order to find a suitable replacement for primaquine, which poses many safety issues [29]. A recent large clinical trial in Mali [10] demonstrated its transmission-blocking efficacy when used as an additional dose (15 mg/kg per day MB for 3 days) in combination with artesunate-amodiaquine for the treatment of uncomplicated *P. falciparum* malaria. MB powerfully reduces the gametocyte carriage rate in treated patients, blocking the transmission of the parasite to the mosquito [10]. However, little information is available on the activity of MB against the subsequent development of *Plasmodium* parasites in the mosquito host. MB has been identified as a non-competitive inhibitor of *P. falciparum* glutathione reductase, which catalyzes the reduction of glutathione disulfide using NADPH as a source of reducing equivalents [16]. Experiments with *P. berghei* indicated that glutathione metabolism is essential for oocyst development in mosquitoes [30]. In terms of its clinical use, MB has been shown to exert highly potent action against asexual *Plasmodium* stages (IC_50_ = 4 nM) [20]. MB also kills gametocytes at all stages, albeit at a somewhat higher rate with stage I-II gametocytes compared to more mature stages [7]. MB has not been shown to possess significant activity against sporozoite motility [31] nor against hepatic stages of *P. falciparum*, *P. yoelii* and *P. cynomolgi* [21]. Here, we provide new evidence of the effects of MB on these stages of the *Plasmodium* lifecycle.

Here we show that the transmission of fresh *P. falciparum* isolates obtained from donors could be disrupted by MB even when added as briefly as only <5 min before membrane feeding. Oocyst development was completely inhibited in all five independent replicates. This result is consistent with previously reported transmission-blocking activity in the murine parasite model *P. yoelii* and *An. stephensi* mosquitoes [21]. Our study may be one of the first to show a robust inhibition of the ex vivo transmission of *P. falciparum* under field conditions and reinforce evidence of using methylene blue as an additional tool to reduce parasite transmission. Unlike ivermectin [32], MB did not affect mosquito survival. Following oral or i.v. MB administration, plasma levels could, however, well achieve the therapeutic doses in humans, and thus, the anti-invasion effect could be clinically relevant [33]. MB also showed a moderate reduction in oocyst numbers and infection prevalence when ingested via a second blood meal to the already *Plasmodium-*infected *Anopheles* mosquitoes 3 days pi. This observation could be explained by the ability of MB to interfere with the glutathione redox system, which is essential for the initiation of oocyst development [30]. Consistent with this model, MB did not significantly reduce the oocyst burden in *An. stephensi* when ingested 6 days after the infectious blood meal. Even though we did not compare MB head-to-head to other drug candidates like ivermectin or atovaquone, the potency of MB appears to be similar or weaker [34,35,36]. Regarding the development of combination therapy, an additional study under similar condition comparing each candidate could of course bring more comparative data.

Exposure of *P. berghei* and *P. falciparum* sporozoites to MB also inhibited the capacity of sporozoites to establish hepatocyte infections. Although MB had previously been shown to have only a limited effect on in vitro sporozoite motility [31], MB seems to affect the capacity of sporozoites for productive hepatocyte invasion. We could hypothesize that MB may induce morphological deformation such as described on gametocytes [22], leading to the incapacity of the sporozoite to invade the hepatocyte. In order to expand on these results, we re-assayed the MB activity against *P. berghei liver* stages at different time point post hepatocyte invasion. We observed a moderate activity of MB against *P. berghei* liver stages, which differs from previously reported results [21]. However, not only the murine parasite species used, as well as the MB formulation provided by our supplier, were different in our study, but also the timeline of the treatment, which was delivered at the same time as the sporozoite inoculation in our study whereas the treatment was initiated 3 h post sporozoite inoculation in the Bosson-Vanga study.

Interestingly, MB had the most pronounced impact when it was added simultaneously to hepatocyte invasion, again suggesting an activity against early phases of hepatocyte infections. MB showed not only an effect on invasion but also on parasite development, as evidenced by the reduction of *P. berghei* EEFs size (Appendix B, Figure A2). The invasion of hepatocytes is not immediate and requires a few hours. Regarding the results obtained from the in vitro sporozoite assay, it is therefore conceivable that MB is rapidly metabolized by hepatocytes and that this metabolization reduces the activity of the parent molecule. This could explain why a higher drug concentration is needed to interfere with the liver-stage development of *Plasmodium* parasites compared to blood-stage parasites, such as those described with other antimalarial drugs [37]. In our in vivo challenge assay, we did not observe a significant inhibitory effect of MB. Although MB showed a significant reduction of parasitemia in mice 6–7 days after infection by mosquito bite, most of the treated mice nonetheless developed blood-stage infections. Even if MB could reduce the liver parasite burden, it failed to completely stop the liver stage invasion and development and, thus, to prevent blood stage infections.

Although some data provided in this study were generated from a single experimental setup, we found similar and consistent results with *P. falciparum and P. berghei*. Our experimental decision was also guided by the 3Rs (Replacement, Reduction and Refinement), which recommends performing single representative experiments as described for other drug candidates [38]. Moreover, our study of the inhibition of mosquito stages by MB lacked a positive control. This was because of a lack of robust data on compounds leading to a complete block of mosquito stage development, as in many previous studies [35,36,39].

## 5. Conclusions

Our study expands our knowledge of the activity of MB as the oldest synthetic anti-infective against the different stages of the complex life cycle of malaria parasites. We demonstrated an extremely rapid activity of MB on *P. falciparum* transmission to *An. gambiae* mosquitoes. Moreover, we found that MB has moderate activity on oocyst development, reducing the oocyst burden and mosquito infectivity. Importantly, exposure of sporozoites to MB revealed a prominent in vitro inhibitory activity of hepatocyte invasion, and we also observed a reduced parasite liver stage load upon exposure of early intra-hepatocyte parasite development to MB in vivo. Facing the emergence and development of malaria resistance, our study thus provides further impetus for the clinical use of MB, for instance, in antimalarial combination therapy.

## Figures and Tables

**Figure 1 pharmaceutics-14-02794-f001:**
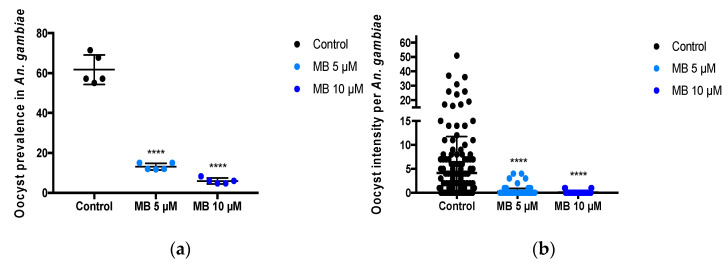
**MB inhibits *P. falciparum* transmission in *An. gambiae*.** (**a**) Prevalence of the infection is presented as the percentage of infected mosquitoes as measured by the presence of oocysts. (**b**) The infection intensity is presented as the number of oocysts per single midgut (black dots). The black lines represent the mean and standard deviation. Data from 5 independent experiments are presented in A–B, corresponding to five different isolates. **** *p* = 0.0001 when compared to control by *t*-test.

**Figure 2 pharmaceutics-14-02794-f002:**
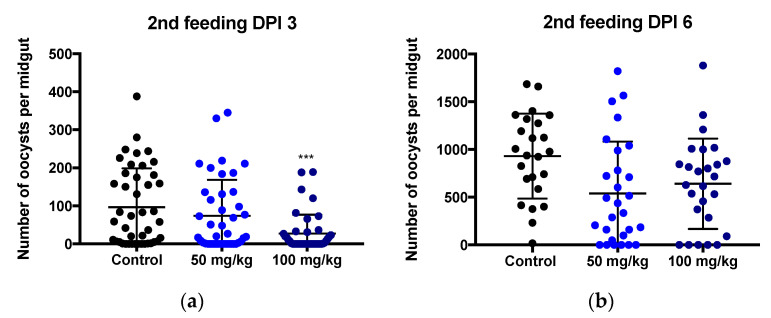
**Exposure to MB during *Plasmodium* mosquito stage development shows a moderate impact on oocyst numbers.** Preliminary *P. berghei*-infected mosquitoes were divided into three groups that received secondary blood meals on anesthetized mice treated with 0 (control, black dot), 50 (light blue dot) and 100 (dark blue dot) mg/kg MB 3 days (**a**) and 6 days (**b**) pi. The infection intensity is presented as the number of oocysts per midgut (black dots). The black lines represent the mean and standard deviation. Data are represented from two independent experiments. *** *p* = 0.005 when compared to the control by *t*-test.

**Figure 3 pharmaceutics-14-02794-f003:**
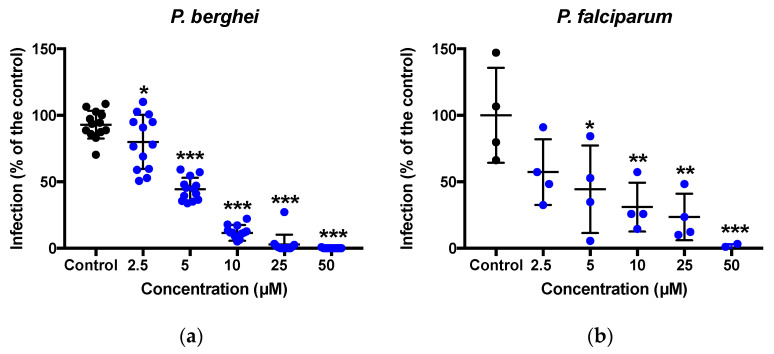
**Exposure of freshly isolated sporozoites to MB inhibits hepatocyte invasion.** (**a**) MB displayed an inhibition (IC_50_ = 4.46 µM) of *P. berghei* hepatocyte invasion after 1 h pre-incubation. Data are presented from four independent experiments. (**b**) MB displayed a potent inhibition (IC_50_ = 4.7 µM) of *P. falciparum* hepatocyte invasion after 1 h pre-incubation. In vitro sporozoite activity (infection scale, blue bars = EEFs). Data are presented from one representative experiment. The black lines represent the mean and standard deviation. * *p* < 0.05; ** *p* < 0.01; *** *p* < 0.001 when compared to control by *t*-test. Control, black dot; MB, blue dot.

**Figure 4 pharmaceutics-14-02794-f004:**
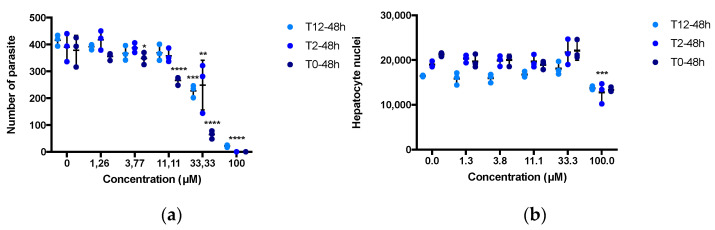
**MB shows weak inhibition of liver stage infection in vitro.** (**a**) MB activity against *P. berghei* liver stages was investigated at concentrations ranging from 1.25 to 100 µM. MB was added simultaneously to the hepatocyte invasion (0 h), 2 h or 12 h pi. Activity is presented as the number of schizonts counted (blue bars = EEFs). (**b**) Toxicity to host cells is presented as the number of hepatocyte nuclei. Data are presented from one representative experiment. The black lines represent the mean and standard deviation. * *p* < 0.05; ** *p* < 0.01; *** *p* < 0.001; **** *p* = 0.0001 when compared to control by *t*-test.

**Figure 5 pharmaceutics-14-02794-f005:**
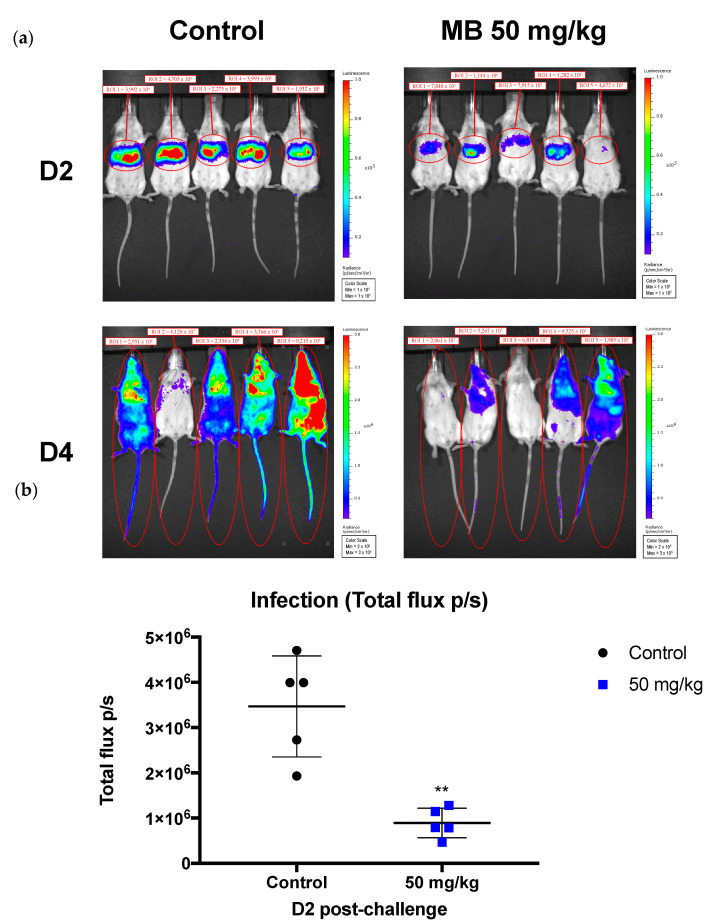
**MB delayed and reduced the *P. yoelii* sporozoite infection in mice.** Mice were first treated for 30 min with MB (50 mg/kg) and were then challenged by retro-orbital injection of 5.000 sporozoites of *P. yoelii.* (**a**) In vivo images (IVIS) of luminescence reported at D2 and D4 post-*P.yoelii*-infection. Rainbow images show the relative levels of luminescence ranging from low (blue) to medium (green) to high (yellow/red). (**b**) Mean luminescence levels (photons/s) for each group at 42 h post-challenge. The black lines represent the mean and standard deviation. ** *p* = 001 when compared to control by *t*-test.

**Figure 6 pharmaceutics-14-02794-f006:**
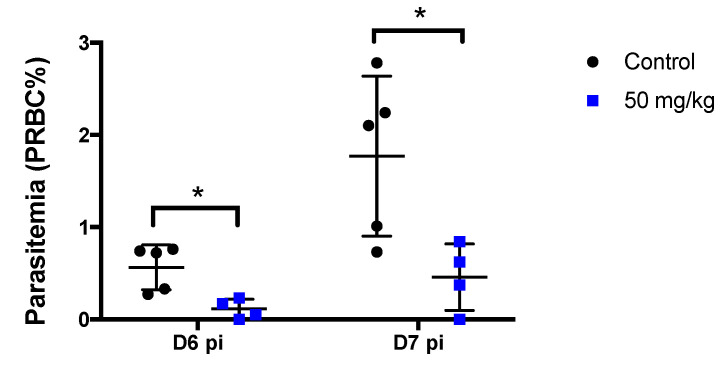
**Single treatment of MB before challenge by mosquito bite reduces subsequent blood-stage parasitemia.** Mice were first treated with MB (50 mg/kg) 30 min before the mosquito challenge and were then challenged by the bite of *P. berghei*-infected mosquitoes. Blood stage parasitemia of *P. berghei* was monitored in mice by studying Giemsa-stained blood smears on day 6 and day 7 after the infection by mosquito bite. Data are presented from one representative experiment. The black lines represent the mean and standard deviation. Asterisks (*) indicate significant (*p* < 0.05) differences between each treatment group and control.

## Data Availability

On reasonable request, the corresponding author will provide the datasets used and analyzed during the current study.

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
