# Peer review of "Rapid and Specific Action of Methylene Blue against Plasmodium Transmission Stages"

_pharmaceutics, 2022, doi:10.3390/pharmaceutics14122794_

Round 1

Reviewer 1 Report

Attached

Reviewer 2 Report

Paper :Rapid and specific action of methylene blue against Plasmodium transmission stages.

The authors investigated the potential transmission blocking ability of Methylene Blue on Plasmodium parasites. They evaluated this by looking at ex vivo direct membrane feeding assays using different Plasmodium falciparum field isolates.  Furthermore, they looked at the effect of MB on the anti-mosquito stage activity by secondary in vivo feeding using Plasmodium berghei-infected Anopheles stephensi mosquitoes. Lastly, they investigated the anti-sporozoite and liver stage effects of MB making use of in vitro and in vivo sporozoite invasion and liver stage development assays, respectively. From this the authors concluded that MB does show inhibitory effects on the transmission of P. falciparum in An. gambiae, in the direct membrane feeding assay, even after short exposure to MB, but limited effect agains the oocyst development. MB was able to block hepatocyte invasion but the development of P. berghei liver stages was unaffected by this compound. These findings are indicating that MB can prevent initial infections in both the human and the mosquito, highlighting again the potential of MB as transmission blocking compound.

General concept comments

  • Is the manuscript clear, relevant for the field and presented in a well-structured manner? 

The manuscript was overall clear and structure in a logical manner, however the readability can be improved.  As malaria transmission is very important for malaria elimination, this paper may contribute to the general knowledge of Methylene Blue in the field and how it can possibly play a role. 

  • Are the cited references mostly recent publications (within the last 5 years) and relevant? Does it include an excessive number of self-citations?

Although the reference list did contain important publications, some of the recent reviews on Methylene Blue in the field was missing, which may be important for the overall conclusion of the paper. (e.g. Bistas E, Sanghavi D. Methylene Blue. 2022 Sep 21. In: StatPearls [Internet]. Treasure Island (FL): StatPearls Publishing; 2022 Jan–. PMID: 32491525)

  • Is the manuscript scientifically sound and is the experimental design appropriate to test the hypothesis?

The manuscript had a clear hypothesis and the experiments proposed were sufficient to answer the aims.  However, the experiments did lack biological repeats and the necessary positive death controls to make sound conclusions. The authors proposed to use MB in similar manner as primaquine and atovaquone is used, so including these as controls may have added validity to the study. Positive controls are important in all experiments, and must be included in these studies. Furthermore, the authors made conclusions based on the morphology, thus it will add value to include examples of these parasite images to show the reader the differences in these parasites. 

  • Are the manuscript’s results reproducible based on the details given in the methods section?

Some details are missing from the methods (such as concentrations used in experiments) and the reader must move to the results and figures to obtain this information. Please include where relevant. Furthermore, the data included in the appendix were not included in the method section.

  • Are the figures/tables/images/schemes appropriate? Do they properly show the data? Are they easy to interpret and understand? Is the data interpreted appropriately and consistently throughout the manuscript? Please include details regarding the statistical analysis or data acquired from specific databases.

The figures were appropriate and they did clearly indicate the data obtained. It may just be of value to include further analysis such as %TRA and TBA in the experiments.

Many of the experiments were only conducted in 1 biological repeat and thus statistical analysis is not possible, which is one of the biggest critiques I have for the paper. In figure 5 the authors did performed stats, which is scientifically inaccurate as this was only performed for one biological repeat. Experiments will need to be repeated to accurately may scientific conductions and conclusions. 

  • Are the conclusions consistent with the evidence and arguments presented?

The overall conclusion was deducted from the claims made in the results, however due to the lack of repeats, the data may not be validated and reproducible.

  • Please evaluate the ethics statements and data availability statements to ensure they are adequate.

All sata is not provided in supplementary but will be made available upon reasonable request. Ethical statement is provided.

Specific comments 

Most comments are made in section above, but overall editorial and language care should be addressed.  Throughout the paper italics are missing, superscript and subscript are used inconsistently, spaces between numbers and units are not consistent and the use of full stop and comma in numbers are also used interchangeable.

Reviewer 3 Report

The study investigated the transmission blocking activity of the well known antimalarial drug, Methylene Blue on the sporogonic cycle of Anopheles mosquitoes. The scientific answers the study provides is important and definitely has relevance to the field, and the work is clearly articulated with sound methodology. 

I feel that the authors could have provided a more clear motivation for the study in terms of how this data will influence policy decisions on the implementation of transmission blocking drugs (ie. through MDA).  Also reference to similar studies using SFMA should also be made, to place the study into context. 

In the introduction, the authors make reference to the MoA of Methylene Blue in context of its activity on metHemoglobin digestion. However, Methylene Blue is a direct inhibitor of GR (Buchholz et al. 2008, and Schirmer et al. 2003 (referenced in discussion), and also a redox cycler drug (Vennerstrom et al. 1995) that produces ROS. I think the authors should include this in their introduction to place the MoA into full context and, make it relatable to the discussion.

The data provided in the study is scientifically sound, although many experiments only present the data from a single biological experiment. Given the technicality of said experiments this is understandable, however, to ensure integrity of the data the authors need to elaborate on this in the statistical analysis section of the manuscript.

Another concern is the lack of reference controls used in the study.  Although the discussion makes reference to a lower activity to Atovaquone based on literature, there is no other reference as to why controls were not used in this study. 

Small typographical considerations include:

1) Consistency in the italization of species names

2) degrees celcius symbol line 129

3) repetitiveness of the following statement in line 144 from the methodology section in line 117

4) reference to figure legend errors in line 225, 278 and 294

5) Use of subscripts for "50" in IC50

6) Miss spel of "mosquitoe" line 316

7) the use of the term "solid" line 359, 380 etc. is a non-descriptive and scientific term, authors could use discernable instead

8) consistent abbreviation of MB

Reviewer 4 Report

The manuscript focuses on additional activities of Methylene Blue (MB) as a transmission-blocking agent and liver-stage infection. The paper is interesting and highlights the efficacy of MB on other life-stages of parasite development. However, this paper would need some revision before being published.

Comment#1: Any experiments to determine whether MB pre-treated host cells (hepatocytes) inhibit invasion by Plasmodium sporozoites?

Comment#2: 

MB shows significant activity against blood-stage infection having IC50 4nM, compared to liver stage having inhibition at sub micro-molar concentration with moderate effect. Why is there differential efficacy in each life stage of parasites? The IC50 values of MB on gametocytes are to be discussed.

Comment#3: The precise/possible mechanistic action of MB against sporozoites and blood-stage parasites is not discussed.

Comemnt#4: Page 12, Line 378 MB is rapidly metabolized by

hepatocytes and that this metabolization reduces the activity of the parent molecule.

How did the authors arrive at this conclusion? the loss of MB activity is correlated with hepatocyte metabolization. How about its plasma half-life?

Comment#5: The authors may comment on whether continuous usage of MB may lead to clinical resistance or MB delays the resistance development compared to standard drugs. Experiments like continuous passage.

Comment#6:

Page7, Line 278 there is a typo error.

The paper's title claims Rapid and specific action of MB, but the findings conclude moderate activity towards oocytes and fail to cure liver stage invasion and development. This completely misleads the readers. When a molecule is claimed rapid, it should be fast-acting and effective inhibition at sub-nano-molar conc.

Why positive controls were not considered in all experiments?

Round 2

Reviewer 1 Report

All reviewer questions and comments have been adequately addressed. I have no further questions or issues with the manuscript.

Author Response

Many thanks for accepting our suggestions and comments addressed throughout the manuscript. 

Sincerely yours, 

Nathanael Saison & Steffen Borrmann 

Reviewer 2 Report

The authors address all issues to the best of their ability and the necessary changes were made.

The manuscript still require editorial work as examples (not limited to):

Line 111: Superscript of the (R) 

Line 132 and 175: thousands are indicated with a full stop (2.000) but line 211 with a comma (5,000). The norm is comma, as decimals are indicated with a full stop. 

Line 153, 208, 240 and 334: min vs minutes, please be consistent.

Line 201: in this single line 2_h_pi and 24h_pi were used - consistency.

Line 199 and 288 as well as 301: spaces not used consistent.

IC50 should be subscript throughout the text. 

Kind regards

Author Response

Sincerely yours, 

Nathanael Saison 

Reviewer 4 Report

The revised manuscript has addressed the concerns raised. The manuscript is acceptable for publication.

Author Response

Many thanks for accepting our comments and suggestions addressed throughout the manuscript. 

Sincerely yours, 

Nathanael Saison & Steffen Borrmann